# Imputation by feature importance (IBFI): A methodology to envelop machine learning method for imputing missing patterns in time series data

**Adil Aslam Mir**[1,2]*, **Kimberlee Jane Kearfott**[3], **Fatih Vehbi Çelebi**[1], **Muhammad Rafique**[4]

**1** Department of Computer Engineering, Ankara Yıldırım Beyazıt University, Ayvalı, Keçiören/Ankara, Turkey, **2** Department of Computer Science and Information Technology, University of Azad Jammu and Kashmir, Muzaffarabad, Azad Kashmir, Pakistan, **3** Department of Nuclear Engineering and Radiological Sciences, University of Michigan, Ann Arbor, Michigan, United States of America, **4** Department of Physics King Abdullah Campus, University of Azad Jammu and Kashmir Muzaffarabad, Azad Kashmir, Pakistan

* adilmir300@gmail.com

**Data Availability Statement:** All relevant data are within the paper and its Supporting Information files.

## Abstract

A new methodology, imputation by feature importance (IBFI), is studied that can be applied to any machine learning method to efficiently fill in any missing or irregularly sampled data. It applies to data missing completely at random (MCAR), missing not at random (MNAR), and missing at random (MAR). IBFI utilizes the feature importance and iteratively imputes missing values using any base learning algorithm. For this work, IBFI is tested on **soil radon gas concentration (SRGC)** data. XGBoost is used as the learning algorithm and missing data are simulated using R for different missingness scenarios. IBFI is based on the physically meaningful assumption that SRGC depends upon environmental parameters such as temperature and relative humidity. This assumption leads to a model obtained from the complete multivariate series where the controls are available by taking the attribute of interest as a response variable. IBFI is tested against other frequently used imputation methods, namely mean, median, mode, predictive mean matching (PMM), and hot-deck procedures. The performance of the different imputation methods was assessed using root mean squared error (RMSE), mean squared log error (MSLE), mean absolute percentage error (MAPE), percent bias (PB), and mean squared error (MSE) statistics. The imputation process requires more attention when multiple variables are missing in different samples, resulting in challenges to machine learning methods because some controls are missing. IBFI appears to have an advantage in such circumstances. For testing IBFI, Radon Time Series Data (RTS) has been used and data was collected from **1st March 2017** to the **11th of May 2018**, including **4** seismic activities that have taken place during the data collection time.

**Funding:** Muhammad Rafique Higher Education Commision, Pakistan Grant No: 6453/AJK/NRPU/ R&D/HEC/2016 under NRPU scheme to principal investigator MR. www.hec.gov.pk The funders had no role in study design, data collection and analysis, decision to publish, or preparation of the manuscript.

**Competing interests:** The authors have declared that no competing interests exist.

# Introduction

Radon ($^{222}$Rn) gas is ubiquitous in the environment. It is found in air, water, and soil, and concentrates in the environment and buildings in a complex manner dependent upon geological, chemical, meteorological, and other temporally variant parameters [1–10]. While the bulk of knowledge about the adverse health effects has resulted from studies of lung cancer in uranium miners, radon health effects are an active area of epidemiological work involving indoor domestic radon gas concentrations [10–14]. Such work often involves indoor radon air concentration time series coupled with data about various multiple environmental and geographic variables. Such data sets may be incomplete, resulting in the need to discard data or perform extrapolations using machine learning or other modeling methods. While radon is of concern when found in hazardously high concentrations in occupied dwellings, it has been found to be beneficial in that it is potentially predictive of earthquakes [15–25]. Various studies show that anomalies in the radon time series data offer strong evidence for earthquake prediction and forecasting [21, 26–30]. Decades of studies have specifically explored the linkages between SRGC and seismic activity [31]. Moreover, soil radon gas emission and transportation dynamics are influenced by various meteorological factors (such as temperature, rainfall, pressure and relative humidity) which are unrelated to seismological activities deeper in the earth crust which also influence radon gas environmental movement [32, 33]. Multiple studies had been performed to analyze the correlation between SRGC and different meteorological parameters [7, 34–38]. A study was conducted at Hokkaido University in Sapporo, Japan for monitoring soil radon gas concentration found that temperature was the dominant meteorological parameter affecting soil radon levels and variability [39]. Sahoo et al. [40] analyzed the influence of meteorological parameter on radon emission dynamics using linear regression analysis. It was observed that temperature is negatively correlated whereas humidity and pressure are positively correlated with radon time series. This study also reports a considerable amount of anomalies prior to the occurrences of local earthquakes with the magnitude of 3.7 and 4.2 Badargadh, India. Different computationally intelligent methods have been proposed and successfully applied to predict radon concentration from environmental parameters such as pressure, rainfall, air and soil temperature [27, 41–44]. Such predictions depend upon data sets which may often include missing information for radon or some of the important environmental parameters which influence its concentrations [45]. This paper concerns itself with a method for imputing, or filling in, missing data to improve the performance of machine learning approaches being considered for identifying seismic abnormalities from soil radon gas concentration (SRGC). Radon health effects and usage of radon as a precursor indication of earthquakes represent prime examples of the interaction of the atmosphere, lithosphere, and hydrosphere with human biology influenced by their behavior and the built environment. Improving the data sets for analysis is the overall goal of this work.

If missing data are not properly imputed this may lead to unreliable outcomes. Within any time series lost data/info may result from human error, instruments failure, or downtime due to routine maintenance purposes [46]. The classification of missing data can be performed by the mechanism through which the missing data is generated [47]. The choice of imputation method is influenced by the actual causes and characteristics of the missing data, whether due to data loss, perceived inapplicability, or lack of relationship to a given situation [48].

The nature of absent data, or missingness, can be classified in three ways[47, 49]. The missing data is said to be completely at random (MCAR) if the probability of the data missing is the same for all the cases, i.e. the missingness of data is not related to the data itself. When the tendency of the data point to be missing is related to the observed data, but not the missing data, then it is called missing at random (MAR). Finally, for data missing not at random

(MNAR), two possible reasons may occur: the missing data point depends on the hypothetical value or the missingness is related to some other variables in the data. To impute missing data, straightforward methods are typically used. Examples are complete or available case analysis, missing-indicator methods, and mean, median and mode imputation. Unfortunately, these approaches may result in severely biased estimates and inefficient analyses [50, 51]. Multiple imputation is a more sophisticated approach to handling missing data that performs better than other conventional methods [52–55]. However, there are certain pitfalls in multiple imputation analyses [56]. When dealing with highly skewed data, multiple imputation results in implausibly low or even negative values. In various scenarios, an analysis needs to explore the association between an outcome and one or more predictor variables, the missing values in the outcome variable result in neglecting the outcome variable in imputation procedure. The omitting of outcome variables would falsely weaken the association among predictors and outcome variables. Moreover, multiple imputation procedure is computationally intensive and some algorithms run repeatedly for better approximation, and its length increases with more missing data. Machine learning methods have been used to reconstruct incomplete and irregularly sampled experimental data for indoor radon gas concentrations [45]. A comparison of traditional statistical and machine learning with available controls methods of data imputation concluded that machine learning outperformed statistical methods and increased the prognosis accuracy significantly [57, 58]. Mital et al. [59] proposed a sequential imputation algorithm for the imputation of missing values in spatio-temporally daily time series precipitation records. The authors demonstrated that the proposed sequential imputation method by incorporating it with a spatial interpolation based on a Random Forest method has several benefits as the number of stations with incomplete records increases. However, the sequential imputation method does not add any extra information for spatial information if the stations having incomplete records decreases. Stochastic semi-parametric regression imputation was found to be superior to existing semi-parametric regression imputation for both simulated and real data [60]. An efficient imputation-based method was also proposed which uses an expectation-maximization (EM) algorithm for multivariate time series data under the assumption of normal distribution [61].

In this study, a more robust methodology for data imputation, Imputation by Feature Importance (IBFI), is proposed and its performance compared with the commonly applied mean, median, mode, hot-deck, and predictive mean matching statistical imputation methods. Actual SRGC data collected over a 14-month period during which time four seismic events occurred is used for the study. Simulations of missing data were made using the R package entitled "mice" [62] for 10, 20, and 30% of the data under MAR, MCAR, and MNAR scenarios. The XGBoost machine learning method was utilized as a base learner for this work. It is noted that any method may be used to impute complex missingness patterns using IBFI, and IBFI may be applied to any machine learning method, such as Random Forest and Naïve Bayes.

## Materials and methods

### Instrumentation and location

SRGC time series data were obtained on the fault line present in Muzaffarabad, a city in the Pakistani territory of Kashmir. The location of the soil radon measuring station is presented in Fig 1. A humidity-insensitive radon and thoron monitor (SARAD RTM 1688–2, Nuclear Instruments, Germany) recorded radon, thoron, temperature, humidity, and barometric pressure at latitude 34.39621 and longitude 73.47347. Readings were integrated over 40 min, resulting in 36 measurements every 24 h for more than 1 y. Additional details concerning the instrument and the resulting data are reported elsewhere [27, 63, 64]. **The statistical details of**

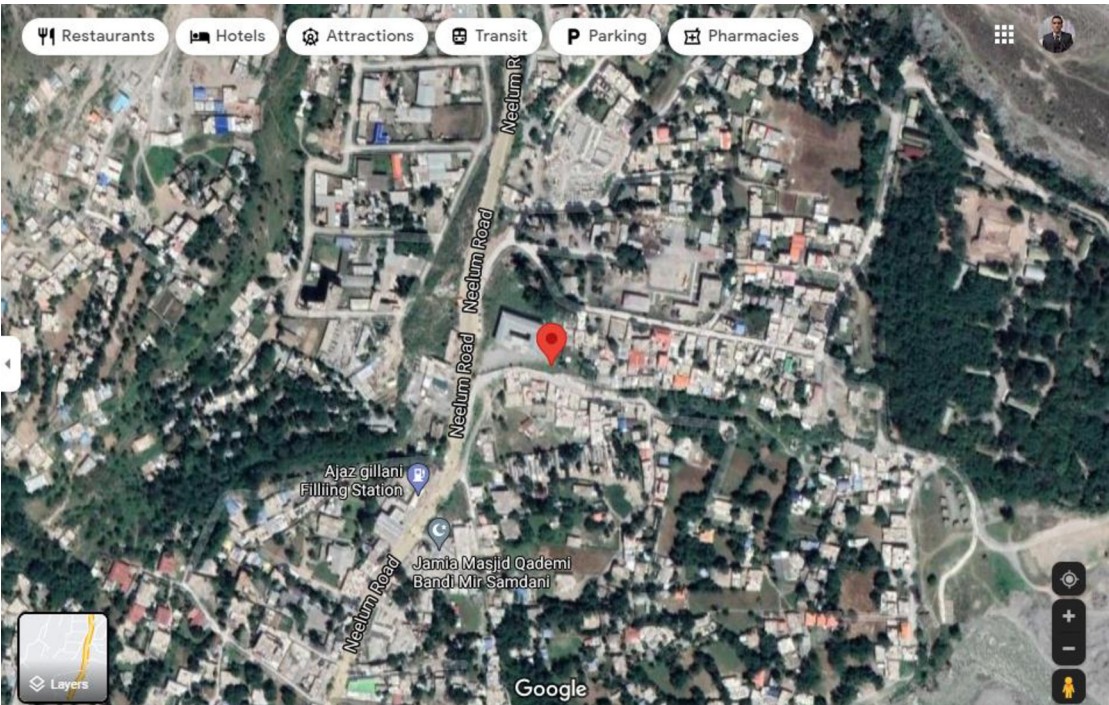

**Fig 1. Soil radon measuring station located inside 150 km from the epicentre of the strongest earthquake since 1900 with the latitude, longitude of 34.39621 and 73.47347 respectively.**

the variables in soil radon gas concentration time series data are provided in **Table 1** below.

The dataset consists of 15692 radon and thoron measurements along with environmental parameters viz. temperature (˚C), relative humidity, and pressure (mbar). Radon concentration (RN) varied from 13743 Bq/m$^3$ to 28085 Bq/m$^3$. Mean and median of radon time series were found to be 21364 Bq/m$^3$ and 21569 Bq/m$^3$ respectively. The temperature varied from 4 to 42.5˚C during the study period.

## Missing data simulation and analysis plan

Fig 2 displays the complete simulation and analysis plan for the current study. The overall SRGC dataset consists of the different attributes (or measured variables) radon, thoron, temperature, relative humidity, and pressure. For the sake of analysis of the imputation methods, the three different missingness patterns (MCAR, MNAR, and MAR) are introduced into the SGRC dataset resulting in modified data sets with 10, 20, and 30% of the data missing. The missing values are introduced into the dataset

**Table 1. Statistical details of the SRGC time series dataset.**

| Variable | Mean | StDev | Minimum | Q1 | Median | Q3 | Maximum | Skewness |
|---|---|---|---|---|---|---|---|---|
| Radon | 21364 | 2130 | 13743 | 19950 | 21569 | 22876 | 28085 | -0.31 |
| Thoron | 2515.3 | 384.3 | 1495.0 | 2246.3 | 2489 | 2761 | 16182 | 4.26 |
| Temperature | 22.485 | 8.085 | 4 | 16 | 23 | 28.5 | 42.5 | -0.05 |
| Relative Humidity | 77.884 | 13.166 | 34 | 70 | 81 | 88 | 101 | -0.81 |
| Pressure | 928.26 | 4.92 | 914 | 925 | 929 | 932 | 943 | -0.28 |

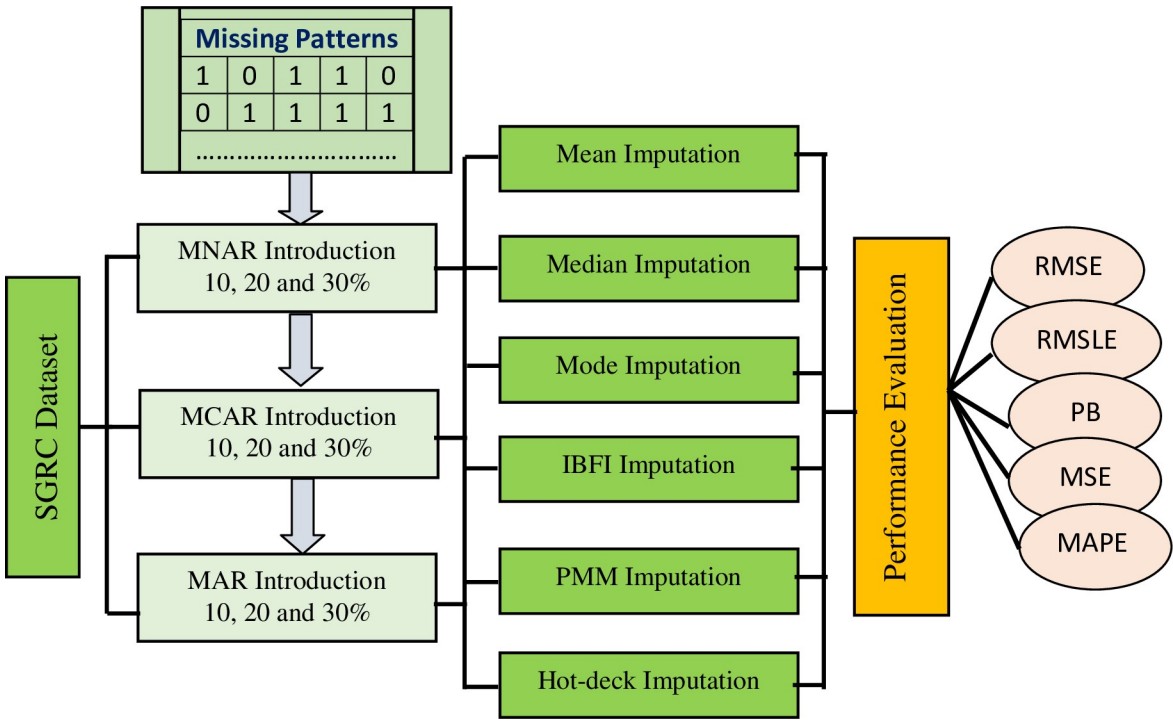

**Fig 2. Simulation plan of the study.**

artificially by the R package entitled "mice" [62]. The core idea for introducing missing values in the multivariate dataset lies in the missing patterns. Where missing patterns are the mixture of variables with missing values and variables with available values [65]. The missing patterns with their frequency are shown in Fig 4. The complete dataset is divided into *k* subsets randomly based upon missing data patterns. The subset size depends upon the frequency vector which is the frequency of the certain pattern to be missing the complete dataset. The data rows in the subsets are considered to be a candidate for missing is based upon several factors such as missingness mechanism (MCAR, MNAR, and MAR). In MCAR scenarios, all the data rows in the subsets have an equal probability of being missing while in MNAR and MAR scenarios, the weighted sum scores are computed. More simply put, the weighted sum scores are the outcome of a linear regression equation. These scores provide the basis for candidates' data rows to be missing or not. Finally, the data rows in the subsets are made missing or incomplete according to the missing data pattern along with its probability of being missing. After the introduction of missing values, these subsets are merged to make an incomplete dataset having missing values in different data rows.

The resulting nine altered SRGC data sets are then treated with six different data imputation methods. These include IBFI and the more common mean, median, mode, predictive mean matching (PMM), and hot-deck imputation methods. Performance metrics computed following the application of the imputation method include root mean squared error (RMSE), mean square error (MSE), root mean squared log error (RMSLE), mean absolute percentage error (MAPE), and percentage bias (PB). The performance of the imputation method is heavily dependent upon the ability of imputation method to impute values that are much nearer to the real value for each of these metrics. Descriptions of both the performance metrics and the imputation methods are given below.

## Performance measures

To assess the performance of imputation models for imputing the missing values of radon, thoron, temperature, relative humidity, and pressure, the following five different statistical parameters are computed: Root mean square error (RMSE), root mean squared log error (RMSLE), mean absolute percentage error (MAPE), mean squared error (MSE), and percentage bias (PB). RMSE is a very frequently used performance evaluation measure for prediction models in many different areas, such as air pollution [66, 67]. This method is sensitive to outliers [68] because each error has an effect on RMSE that is proportional to the size of the squared error and thus larger difference between actual predicted value results in an excessively larger effect on it. RMSE is the square root of the average of squared errors computed over a total number of values T, specifically:

$$RMSE = \sqrt{\frac{1}{T}\sum_{k=1}^{T}\left(Predicted_k - Actual_k\right)^2} \tag{1}$$

The RMSLE is obtained from the log of predicted and observed values, namely:

$$RMSLE = \sqrt{\frac{1}{T}\sum_{k=1}^{T}(\log(\text{Preds}_k + 1) - (\log(\text{Act}_k + 1))^2} \tag{2}$$

The RMSLE is employed when it is desirable to avoid over-penalizing huge differences in the predicted and observed values in the case when those values are very high numbers. While RMSE is sensitive to outliers and explodes the error term when these are present, RMSLE seriously scales down the impact of outliers. It should be noted that RMSLE penalizes the underestimation of the observed values more severely than it does for overestimation. The MAPE is a frequently used statistical measure of how accurate a prediction system is, computed from:

$$MAPE = \frac{1}{T}\sum_{k=1}^{T}\left|\frac{Actual_k - Predicted_k}{Actual_k}\right| \tag{3}$$

The principal advantage of expressing the MAPE as a percentage, as opposed to simply reporting the mean absolute error, is that it is easier for researchers to conceptualize. The weakness arising from the normalization is that the MAPE becomes undefined datasets that contain values of 0.

The Mean Squared Error (MSE) is a measure that finds out how much close the predicted and observed values are, and is given by:

$$MSE = \frac{1}{T}\sum_{k=1}^{T}\left(Actual_k - Predicted_k\right)^2 \tag{4}$$

For each predicted value, the distance is measured from the corresponding actual value and then squares the resultant value. More simply put, the metric is the average of the squares of errors. The average tendency of the predicted value to be smaller or larger than that of its actual value is captured by the PB performance metric, defined as:

$$PB = 100 \times \frac{\sum_{k=1}^{T}\left(Predicted_k - Actual_k\right)}{\sum_{k=1}^{T}Actual_k} \tag{5}$$

A PB of 0 is considered to be an optimal value indicating accurate model simulation with values having low magnitude. Larger positive and negative values indicate overestimation and underestimation bias, respectively.

## Mean, median, and mode imputation methods

The mean model for imputation is a method in which the mean of the observed cases (all non-missing values of the attribute of interest) of the certain variable serves as a replacement for missing values in that variable. The simple-to-use mean model inherently reduces the variability in the data, resulting in an underestimation of standard deviation and variance estimates. Median imputation substitutes the middlemost number in the observed values when these are arranged in order. Mode imputation replaces missing data with the most frequently occurring value for that particular variable. SRGC data consist of attributes that are continuous, meaning that no two values will be the same exactly. For this reason, for mode imputation kernel density estimation is used to produce a continuous estimate of the probability density function. The point at which the probability density function reaches a maximum is considered its mode. For kernel density estimation, R package "stats" [69] has been used. The function "density ()" with its default parameters are used to calculate the kernel density estimate. The function "density.default ()" uses the algorithm that first use a regular grid of at least 512 points to disperse the mass of the empirical distribution function. The fast Fourier Transform along with the discretized version of the kernel such as Gaussian is used to convolve the approximation. Finally, the densities at specified points are evaluated using linear approximation.

## Hot deck imputation method

Hot deck imputation is the method to impute missing data of one or more features for a non-respondent, called the recipient, where each missing value is substituted with a practical response from a "similar" unit i.e. it involves replacing the targeted missing values with those from a "similar" responding unit (the donor). Though, Hot-deck imputation is an old but popular method of imputation because it is simple in concept and suitable for missing at random (MAR) patterns. The basic principle is to locate one appropriate donor value from the available observed case that is comparable to the missing case in some regards [70]. The donor is similar to the recipient for features observed in both cases. The random hot-deck imputation method involves the random selection of donors or respondents from a set of possible available donors called the donor pool. There are other versions of this method involving a single donor and values are replaced from that case, generally, the "nearest neighbor" based on some metric; these methods are called deterministic hot-deck methods as no randomness is involved in the donor selection. However, the hot deck imputation has certain limitations e.g., good matches of respondents or donors are required by it to recipients reflecting available covariate information. There are cases when the single donor may be chosen to accommodate several recipients' leads to replication of values [71]. This replication of values causes several problems and there is an inherent risk that lot of missing values or even all of the missing values gets imputed from a single donor. The hot-deck method does not take the correlation of the variables into account when imputing values in different features. The imputation procedure is univariate and does not distinguish the multivariate nature of the dependent variables. Due to the copying or borrowing of value from the available case, another problem that arises when imputing with hot-deck imputation is the addition of random noise if the value is quantitative. The missing values were imputed through Hot-Deck using the R language package entitled "VIM" [72].

## Predictive mean matching (PMM) imputation method

The predictive mean matching (PMM) method existed a long time ago [73, 74], but its widespread and practical applications began only recently. For the multiple imputation of the missing data, predictive mean matching (PMM) [73, 75] is considered a good method, typically

when the quantitative features are involved that are not normally distributed. It is the state-of-the-art hot deck multiple imputation method [76]. The imputed values will be skewed, if the original feature values are skewed and bounded by some upper and lower limit e.g., 0 to 50 if the original feature is bounded by the limit. The reason is that imputed values are the original values that are borrowed from individuals with original data. The potential donees and donors, selected by either automatic distance-aided or nearest neighbor method, are matched in PMM by the closeness of predicted means. Considering each donor case, the predicted value for the incomplete case is compared to the fitted value obtained from some regression model. More-over, in the classical PMM approach, a case is drawn from the pool of $k$ cases whose estimated values are nearer to one of the value predicted for the missing case. Further, the missing value is imputed by the observed value of donor case. Initially, it was limited in usage i.e. only a single variable with missing data could be handled by PMM or, more broadly, its applications were limited to the situations where there existed monotonic missing data patterns. The PMM method has been embedded in various software packages that employ multiple imputation approaches, referred to as sequential generalized regression (SGR), fully conditional specification (FCS), or multiple imputation by chained equations (MICE). The quality of imputed values depends upon the availability of appropriate donor cases. In small datasets, the imputation by predictive mean matching could not give promising results, as there might not be suitable donor cases available.

In the current study, missing values were imputed through PMM using the R language package entitled "mice" [62] with the parameters i.e. m (stands for 'number of multiple imputations'), maxit (stands for 'no of iterations'), method, and the seed of 5, 500, pmm, and 50 respectively. The 'm' with a value of 5 (considered to be enough [75] and also a default value) will generate five imputed datasets that differ only in imputed missing values. In classification or regression problems, the prediction models build upon these imputed datasets perform better by aggregating the prediction of these models. Considering the importance of the imputation process to reach convergence, a maximum number of iterations have been chosen i.e. 500. Generally, in the region of 20 to 30 or fewer iterations for each imputation are taken as a rule of thumb. Also, a random seed value of 50 is chosen for reproducibility.

## Imputation by feature importance (IBFI) method

A pictorial representation of this new imputation method appears as Fig 3, while the coded algorithm itself appears in pseudo-code format below. The proposed method starts with the input data matrix (DM) which contains the different attributes (or quantities), specifically SRGC, thoron concentration, soil temperature, pressure, and humidity. DM contains different types of missingness (MNAR, MCAR, and MAR) of values of the attributes shown in Fig 2.

**Pseudocode:** as implemented, for the imputation by feature importance (IBFI) method.

```
Input:
DM(X₁...Xₙ) = Data Matrix where X₁...X_n contains the missing values
RejectionThreshold = The extent to which the number of features in
each sample gets imputed
FIM = Feature Importance Matrix having feature importance for each
feature in descending order
BM = Base Model
Set
ModelList = NULL
U ← sequence(1 to number of features)
PriorityMat ← NULL
k = 1
Process:
```

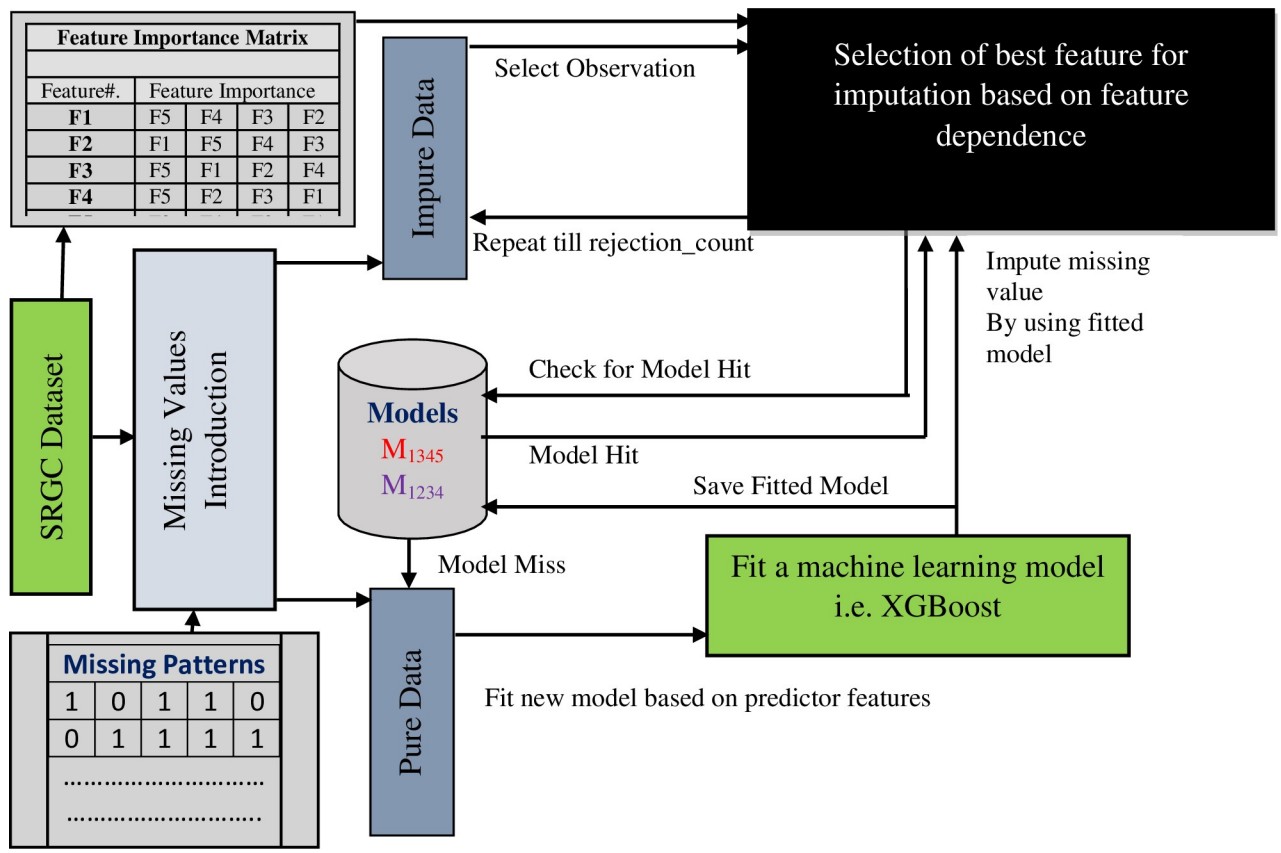

**Fig 3. Proposed methodology to envelop base learning algorithm for imputation.**

1. Split **DM** in to two parts i.e. PD(X₁...Xₙ) and ID(X₁...Xₙ) **//Pure data (PD) has feature values available for all datarows whilst Impure data (ID) has one or more missing values**

2. **while** k < = RejectionThreshold

   a. **For** scanindex = 1 to number of rows in **ID**

      i. NAvector = indices(is.NA(**ID**[scanindex,] == **TRUE**)) **// To the find features having missing values at certain scan index**

      ii. **If** length(NAvector)> **RejectionThreshold** OR length (NAvector) == 0)

         1. **Reject that sample**

      iii.**Else**

         1. TrainVector = **setdiff**(U,NAvector) **// List of features other than having missing values**

         2. Priority_Mat = **FIM**[NAvector,] **// feature importance vectors of the missing features**

3. For all the indices where Priority_Mat contain values of TrainVector, Set Priority_Mat[indices] = 0

4. For all the indices where Priority_Mat do not contain values of TrainVector, Set Priority_Mat[indices] = 1

5. **For** n in 1 to RowsCount(Priority_Mat))

 a. **For** j in 1 to ColumnsCount (Priority_Mat))

 i. if(Priority_Mat[n,j] == 1)

 1. location_vector[l_c] = j

 2. l_c = l_c + 1

 3. break

6. **IF** length(location_vector) == 0)

 a. indice_to_train = NA_vect

7. **Else**

 a. max_value = max (location_vector)

 b. max_i = indices where (location_vector == max_value)

 c. max_i = max_i[1]

 d. indice_to_train = NAvector[max_i]

8. **End if**

9. Traindata = **PD**[,TrainVector]

10. Trainclass = **PD**[,indice_to_train]

11. trainF = Concatenate Column(traindata,class = train-class) // to concatenate response variable with predictors

12. ModelName = concatenate(indice_to_train, TrainVector)

13. Flag = **CheckModel**(ModelList,ModelName) // check for model reusability

14. **IF** flag == -1 // Model not in the already fitted ModelList

 a. Fit a machine learning model i.e. $BM_{MName}$ and save the fitted model to ModelList

 b. Testdata = **ID**[scanindex,TrainVector]

 c. Val = predict(model,testdata)

 d. **ID**[scanindex,indice_to_train] = val

15. **Else**

 a. print("Model Hit")

```
    b. testdata = ID[scanindex,TrainVector]

    c. val = predict(Model_List[[flag]],testdata)

    d. ID[scanindex,indice_to_train] = val

16. End if

iv. End if

3. k = k + 1

4. End

Procedure: CheckModel (ModelList, ModelName):

1. Flag = -1

2. For i in 1 to length(Model_List)

 a. IF ModelList[i] == ModelName

    i.  Flag = 1

    ii. break

3.  Return flag
```

In the very first step, the original DM is divided into two subsets: Pure Data (PD) and Impure Data (ID). PD contains those samples from the data matrix which do not have any missing values, while ID consists of those samples that had between one and a maximum of $n$ missing values per sample. The IBFI uses Pure Data (PD) to fit a machine learning model for imputing missing values in Impure Data (ID). As values are imputed during the IBFI process, predicted missing values are continuously imputed to their respective locations and during iterations, completing the missing patterns based on the feature importance matrix (FIM). The feature importance matrix is computed by using the R package "randomForest" [77]. For each feature in the pure dataset (PD) by taking it as a response feature and others as the predictor, the feature importance of other features for predicting that response is computed. The features are arranged as per their importance value from highest to lowest. The imputation of missing values for a given attribute is next executed on the ID. This is done by applying a machine learning model trained using the non-missing attributes in pure data (PD) to predict the value in missing attributes of impure data (ID). Consider the different attributes $F_1, F_2 \ldots F_n$. If the missing value occurs in $F_1$, the remaining attributes $F_2 \ldots F_n$ will be used for training any machine learning model and the resulting fitted model will be used to predict the missing values for $F_1$. If the attribute $F_3$, is missing, then the attributes $F_1, F_2, F_4 \ldots F_n$ will be used for training any machine learning model and $F_3$ will be predicted from that fitted model. These predicted values will serve in place of the missing values.

The complexity increases when a sample has more than one missing value and certain features or attributes exhibit strong dependencies. Suppose DM contains the five attributes $F_1, F_2, F_3, F_4, F_5$ and missing values occur in $F_1$ and $F_5$ of some samples as shown in Fig 5. Also, assume that certain attributes have a strong correlation with other attributes in the DM. For example, suppose $F_1$ and $F_5$ have a feature importance vector with other attributes from highest to lowest of $F_5, F_3, F_4, F_2$ and $F_2, F_4, F_1, F_3$ respectively. For those samples having $F_1$ *and* $F_5$ as missing values, conventionally $F_2, F_3, F_4, F_5$ and $F_1, F_2, F_3, F_4$ will be used for training the model. The imputation process becomes complicated for a machine learning model to impute the values for $F_1$ and $F_5$ when both have missing values in different samples. In this scenario,

$F_2$, $F_3$, $F_4$ are only attributes available for training the model because predicting the missing value of $F_5$ needs the available value for $F_1$ and vice versa. Moreover, to efficiently impute the value for $F_1$ *and* $F_5$, one must decide whether $F_1$ or $F_5$ must be imputed first. From the correlation vector above $F_5$ is the most important attribute to predict the value of $F_1$ whilst $F_2$ is the most important attribute to predict the value of $F_5$. Based on feature importance $F_5$ needs to be imputed first and when the value of $F_5$ is available then that value of $F_5$ will be used to predict the value of $F_1$. Initially, there may be missing values for multiple attributes. The best attribute to impute is first selected, and the missing values for that attribute are imputed. After this is completed, there is now one less attribute that has missing values. The best attribute to impute after that first process is completed is then determined, and its values are imputed. The process continues until there are no attributes that contain missing values.

The order of selection of attributes for imputation is determined by the FIM, which is determined by calculating the variable importance for each attribute of the data set and arranging the values in descending order. Because it is an iterative approach, IBFI requires a termination criterion. For this purpose, the number of missing values per sample, termed the rejection threshold, is selected. As Fig 3 shows there exist multiple features are missing at once in a sample e.g. $F1$ and $F4$, the rejection threshold is the extent to which we want to impute the number of missing values per sample. If the rejection threshold is 3 it means that if the number of missing values per sample is greater than 3, those samples will be rejected, and all the other samples get imputed in their respective scans. The methodology works in such a way that during the first iteration when keeping the rejection threshold of 3; all the samples having missing values of count 3 will be reduced to missing at 2 per sample as shown in Fig 4. After the second iteration, missing at 2 per sample will be reduced to missing at one per sample and further missing at one per sample is imputed and we got full imputed data other than those samples whose missingness count lies above the rejection threshold. Moreover, the proposed methodology uses model reusability to make it asymptotically better by storing the models which are fitted during subsequent iterations. The models are stored in such a way that if $F1$ is the dependent feature while $F2$ and $F3$ are independent features then model is stored in the memory as $M_{123}$. In the subsequent iterations e.g. missing at 3 features is reduced to missing at 2 features and

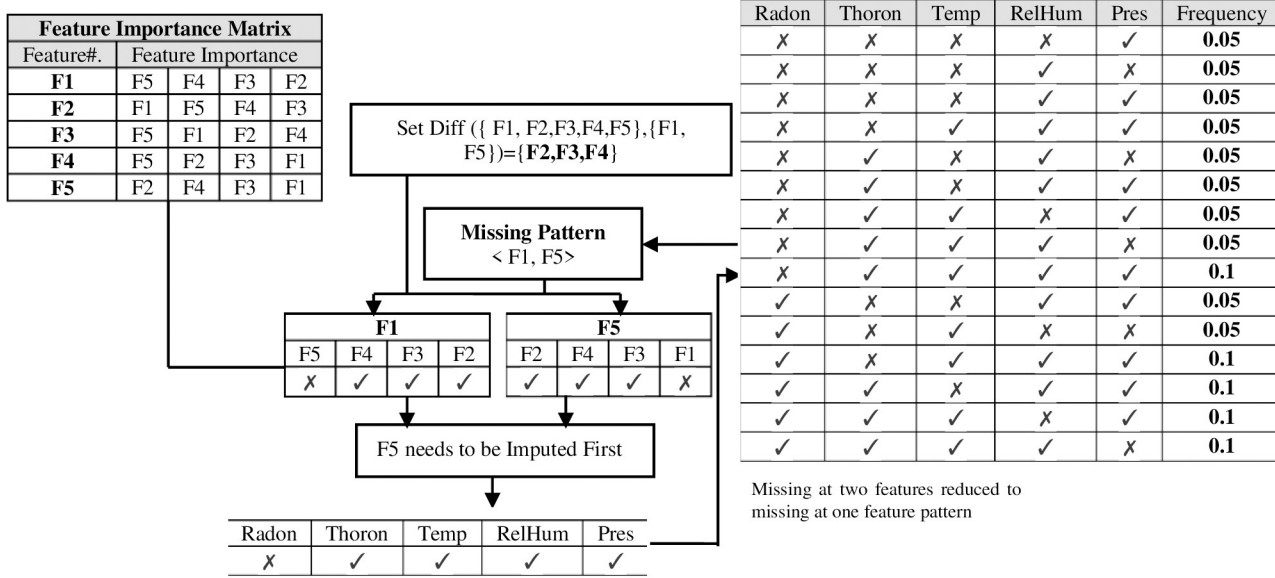

**Fig 4. Proposed methodology to select a feature needed to imputed first in imputation by feature importance (IBFI) method.**

again F1 needs to be trained using *F*2 and *F*3; instead of fitting another model the same model $M_{123}$ will be used to impute the value for *F*1. The sequence of imputation of values for the missing features in different samples differs from each other. The whole procedure is scanning the impure dataset as per the rejection threshold. Consider data row 1, the features *F*1 and *F*5 having a missing value while data row 100 have missing values at features *F*1, *F*2, and *F*5. Data row 1 has F2, F3, and F4 feature values available to predict the value of F1 and F5 whilst F3 and F4 have only available feature values to predict the missing value of F1, F2, and F5. During scan 1 and iteration 1, the feature importance matrix directs the algorithm to predict the missing value of F1 using the model fitted on features F2, F3, and F4 by considering F1 as a response and F2, F3, and F4 as predictor features. After predicting and imputing the missing value of F1, the model should store as $M_{1234}$ for future patterns. Through the same iteration when scanning data row 100, the feature importance matrix directs the algorithm to impute the value of F2 which is missing by taking F2 as a response attribute while F3 and F4 as predictor attributes. After predicting and imputing the missing value of F2, the data row 100 has now an available value of F2 to predict other features. Now consider the second iteration and scan 100, the feature importance matrix directs the algorithm to impute the value of F1 first instead of F5. Currently, F2, F3, and F4 are available values to predict the missing value of F1. Instead of fitting another model to predict the missing value of F1, the previously generated $M_{1234}$ model is reused and values of F2, F3, and F4 are passed to that model to predict the missing value of F1. The reusability of the already fitted model to impute the similar missing patterns enhance the performance and reduces the time and space requirements.

## Results and discussion

The MAPE and PB statistics for all methods tested are shown in Tables 2 and 3 for all variables, radon (RN), thoron (TH), temperature (TC), relative humidity (RH), and pressure (PR), for 20% MCAR, MNAR, and MAR data. All methods had <0.03% MAPE for PR, expected as pressure variations are generally very small and regular in time. RN and TH had similar statistics for a given method, with IBFI performing the best compared to all other methods. As shown in this table, for 20% missingness the MAPE for IBFI for RN ranged between 0.50 and 0.53%, with this statistic being up to 1.8 times higher for Hotdeck. IBFI is similarly superior to all other methods for imputing TC values. The average MAPE for 20% MCAR was 0.8%

**Table 2. MAPE and PB statistics for IBFI compared with other imputation methods (mean, median, mode, PMM, and Hotdeck) for 20% missingness of type MCAR and MAR and all parameters tested (RN, TH, TC, RH, and PR).**

| Method | Statistics | MCAR 20% | | | | | MNAR 20% | | | | |
|---|---|---|---|---|---|---|---|---|---|---|---|
| | | RN | TH | TC | RH | PR | RN | TH | TC | RH | PR |
| IBFI | MAPE | 0.53% | 0.48% | 0.83% | 0.24% | 0.01% | 0.52% | 0.46% | 0.54% | 0.18% | 0.01% |
| IBFI | PB | -0.04% | -0.05% | -0.18% | -0.01% | 0.00% | 0.31% | 0.21% | 0.12% | 0.07% | 0.01% |
| Mean | MAPE | 0.69% | 0.72% | 2.78% | 0.74% | 0.02% | 0.67% | 0.75% | 1.71% | 0.47% | 0.02% |
| Mean | PB | -0.05% | -0.11% | -1.38% | -0.18% | 0.00% | 0.42% | 0.42% | 0.46% | 0.34% | 0.01% |
| Median | MAPE | 0.69% | 0.71% | 2.84% | 0.75% | 0.02% | 0.63% | 0.77% | 1.70% | 0.37% | 0.02% |
| Median | PB | -0.13% | -0.05% | -1.56% | -0.36% | 0.00% | 0.35% | 0.47% | 0.38% | 0.18% | 0.01% |
| Mode | MAPE | 0.72% | 1.07% | 3.01% | 0.75% | 0.03% | 0.86% | 1.42% | 2.91% | 0.30% | 0.03% |
| Mode | PB | 0.17% | 0.98% | 2.61% | -0.18% | 0.02% | 073% | 1.40% | 2.71% | -0.14% | 0.03% |
| PMM | MAPE | 0.81% | 0.71% | 1.28% | 0.33% | 0.02% | 0.77% | 0.67% | 0.82% | 0.26% | 0.02% |
| PMM | PB | -0.09% | -0.11% | -0.24% | -0.01% | 0.00% | 0.32% | 0.21% | 0.10% | 0.08% | 0.01% |
| Hotdeck | MAPE | 0.95% | 0.98% | 3.47% | 0.96% | 0.03% | 0.88% | 0.94% | 2.24% | 0.64% | 0.03% |
| Hotdeck | PB | -0.02% | -0.14% | -1.34% | -0.13% | 0.00% | 0.45% | 0.42% | 0.40% | 0.34% | 0.01% |

**Table 3. MAPE and PB statistics for IBFI compared with other imputation methods (mean, median, mode, PMM, and Hotdeck) for 20% missingness of type MAR and all parameters tested (RN, TH, TC, RH, and PR).**

| Method | Statistics | MAR 20% | | | | |
|---|---|---|---|---|---|---|
| | | RN | TH | TC | RH | PR |
| IBFI | MAPE | 0.50% | 0.48% | 0.97% | 0.23% | 0.01% |
| IBFI | PB | 0.01% | -0.03% | -0.16% | -0.01% | 0.00% |
| Mean | MAPE | 0.65% | 0.74% | 3.08% | 0.63% | 0.02% |
| Mean | PB | 0.17% | -0.15% | -2.28% | -0.24% | -0.01% |
| Median | MAPE | 0.63% | 0.73% | 3.14% | 0.66% | 0.02% |
| Median | PB | 0.09% | -0.09% | -2.37% | -0.41% | -0.01% |
| Mode | MAPE | 0.75% | 0.71% | 2.82% | 0.93% | 0.02% |
| Mode | PB | 0.44% | 0.10% | 2.38% | -0.90% | 0.01% |
| PMM | MAPE | 0.78% | 0.68% | 1.39% | 0.36% | 0.02% |
| PMM | PB | -0.02% | -0.06% | -0.28% | -0.03% | 0.00% |
| Hotdeck | MAPE | 0.92% | 1.00% | 3.93% | 0.83% | 0.03% |
| Hotdeck | PB | 0.19% | -0.14% | -2.34% | -0.26% | -0.01% |

compared to 1.3–3.5% for other methods. For TC, and 20% MNAR, IBFI had a MAPE of 0.5% compared to 0.8% to 2.9% for other methods. The 20% MAR data has similar results, namely 1% for IBFI compared to 1.4–3.9% MAPE for other methods. When PB is considered, the absolute percent bias for IBFI was lower than for all other methods and variables for 20% MNAR, MCAR, and MAR data with 20% missingness.

Fig 5 shows the results when the statistics are normalized to the same statistic averaged over all of the methods, then averaged across all variables. The average statistics are calculated for all variables across different missingness scenarios such as MCAR, MNAR, and MAR. Firstly, the average of each performance metric among all imputation methods is calculated for the entire variables across different missingness scenarios with their associated missingness percentages. Secondly, each value in different missingness scenarios such as MCAR 10% is normalized with respect to different performance metrics by dividing with the corresponding average value calculated in step 1. This results in normalized values of each performance metric across different missingness scenarios for different missingness percentages. Thirdly, the averages of different metrics for all the variables are calculated with respect to imputation method. As can be observed in Fig 5, there is little difference as a function of data missingness type and degree for the RMSE, RMSLE, MAPE, and MSE statistics. IBFI is superior for all cases for these statistics. The percent bias (PB), as illustrated in Fig 6, appears to be somewhat dependent upon both the type and degree of missingness. For MCAR data, the PB for IBFI is similar to Hotdeck and PMM for 10% missingness, but superior for greater degrees of missingness (20%, 30%). With MNAR data, PB shows only positive bias. IBFI is similar to PMM for all degrees of missingness but better than the other methods for these. When missingness is MAR, mixtures of negative and positive PB are observed. IBFI has a similar PB to Hotdeck and Mean at 10% MAR, and to PMM at 10% and 20% MAR. For Mode >10% MAR missingness, IBFI shows lower PB than all methods.

The complete statistical results (RMSE, RMSLE, MAPE, MSE, PB) for IBFI compared with other methods (mean, median, mode, PMM, Hotdeck) for different types (MCAR, MNAR, MAR) and degrees (10%, 20%, 30%) of data missingness are provided in the supplementary materials.

Fig 7A–7C shows the previously fitted model reusability in subsequent scans at missing completely at random with different missingness percentages ranging from 10 to 30 percent.

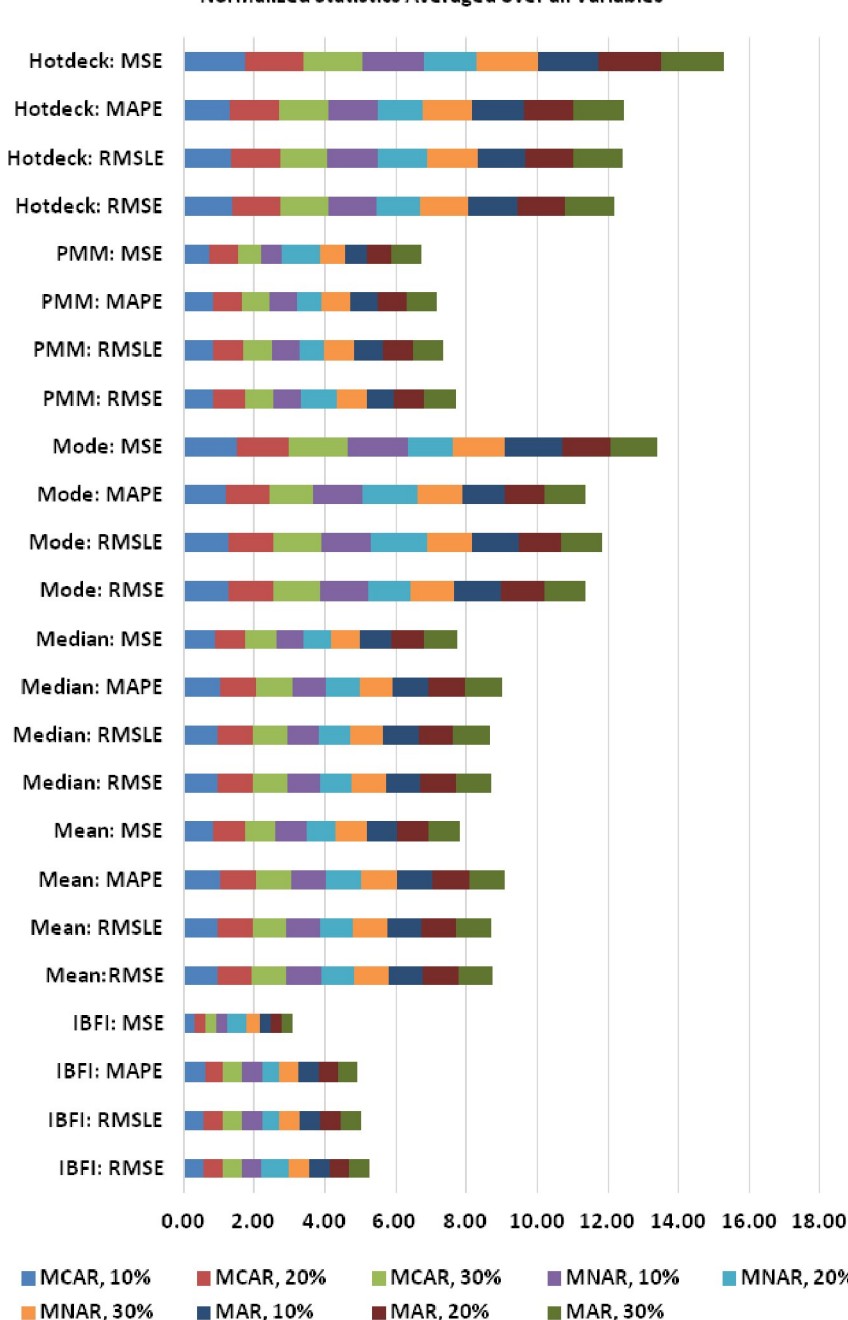

**Fig 5. Statistics for IBFI compared with other methods (mean, median, mode, PMM, Hotdeck) normalized to the average statistic for all methods averaged across different variables, showing RMSE, RMSLE, MAPE, and MSE.**

On the X-axis there is a sample number in subsequent iterations while model hit rate and model creation is shown on the Y-axis. The proposed methodology uses model reusability by keeping the models which are fitted during subsequent iterations for future patterns. Fig 7A shows that with the processing of samples, fitted models are stored and in the subsequent samples those models are utilized that is represented with the black dotted line. Fig 7A–7C shows that with the advancement of samples the model hit rate increases rapidly and is shown in

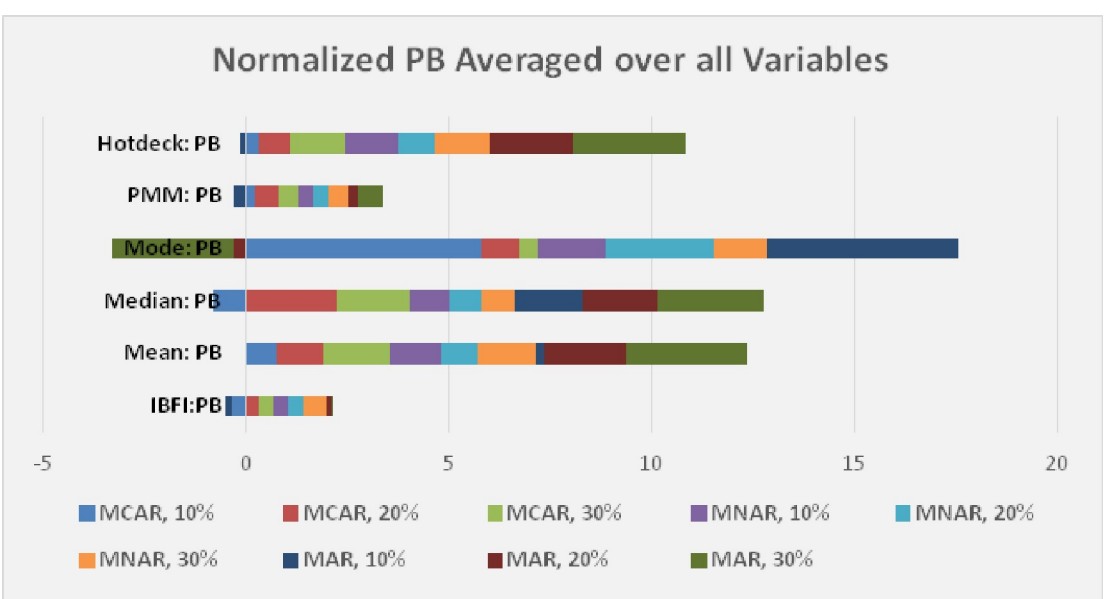

**Fig 6. Statistics for IBFI compared with other methods (mean, median, mode, PMM, Hotdeck) normalized to the average statistic for all methods averaged across different variables, showing PB for 10%, 20%, and 30% MCAR, MNAR, and MAR missingness.**

black dotted line. Model hit reflects the previous model's reusability. While imputing each value, before model creation, the current formulated model directed by the feature importance matrix is searched in the model list. If it is found, the already fitted model should serve to predict the missing value at this stage. In the case when the model hit does not occur, the new model is fitted for the current formulation, and the fitted model is added to the model list which may use for further formulations. The point where the new model is created is shown as blue bubbles. As shown the model hit rate of the proposed methodology is getting higher with the advancement in the processing of samples. The model creation is just observed during the first few measurements and further that models were used for the prediction of missing values in upcoming samples. The reusability of models in such a way helps the proposed methodology to impute the missing patterns asymptotically better in terms of time and space. The same pattern was observed in Fig 7D–7I which shows the fitted model's reusability statistics during the subsequent iterations (missing not random, missing at random). The model creation was observed just at the start of the imputation process as shown in red bubbles but the hit rate of these fitted models increases with the advancement of measurements. This reusability results in the efficient imputation of missing values for all the missing scenarios and makes it asymptotically better.

## Conclusion

Real-time series often contain missing values and missingness can arise for many possible reasons. The situation becomes very important when missingness induces bias in the forecasting model. In this article a methodology has been proposed that utilizes the feature importance and iteratively imputes the missing values in the time series data by incorporating any machine learning model e.g. XGBoost. The proposed methodology imputes various complex patterns of missingness and sets the rejection count that automatically rejects those samples whose number of missing values matches the count. Missing values patterns in the data have been

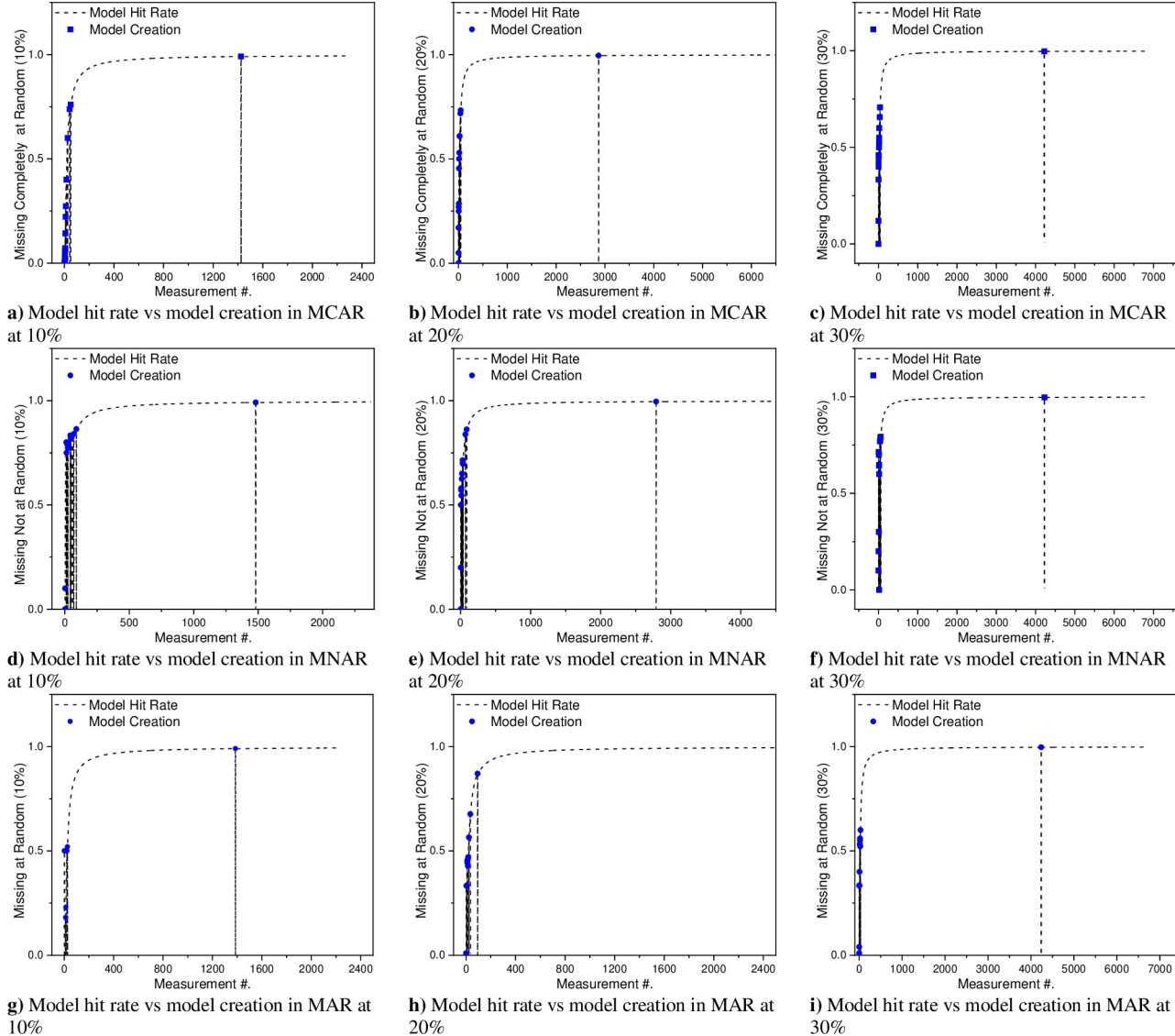

**Fig 7.** Previously fitted model reusability in subsequent scans with different missingness percentages ranging from 10 to 30 percent, for data missing, with data: a/b/c) completely at random, d/e/f) not at random, and g/h/i) at random.

simulated at different missing percentages ranging from 10 to 30 percent in terms of missing completely at random (MCAR), missing not at random (MNAR) and missing at random (MAR) scenarios. In this way, artificially missing value patterns have been introduced in different features. On imputing the same incomplete data, the proposed methodology outperforms than other frequently used methods such as mean, median, mode, predictive mean matching, and hot-deck imputation. Different statistical parameters, viz. RMSE, RMSLE, MAPE, and MSE, have been calculated and indicates that the proposed methodology-based results got very less error values when compared to other imputation methods at different missing scenarios of MCAR, MNAR, and MAR with the percentages of 10, 20, and 30 percent. The findings of the study show that the efficiency of the proposed methodology lies in the selection of the best predictor variable for different missingness patterns and the utilization of previously fitted models. The runtime decision of choosing the best and available predictor variables for different

response variables results in the efficient development of machine learning model for imputing the values. As far as future directions are of concern, the application of the proposed methodology to other fields of research may be of interest such as electric load forecasting and medical databases. Imputation by feature importance (IBFI) can be extended to add class information while imputing supervised classification datasets.

## Supporting information

**S1 Table.** Summary of the results of the simulations at missing completely at random (MCAR) with the missingness percentage of a) 10%, b) 20%, and c) 30.
(DOCX)

**S2 Table.** Summary of the results of the simulations at missing not at random (MNAR) with the missingness percentage of a) 10%, b) 20%, and c) 30.
(DOCX)

**S3 Table.** Summary of the results of the simulations at missing at random (MAR) with the missingness percentage of a) 10%, b) 20%, and c) 30.
(DOCX)

## Acknowledgments

We are thankful to Mr. Aleem Dad Khan Tareen, PhD scholar at the Department of Physics University of Azad Jammu and Kashmir for helping in the process of data acquisition.

## Author Contributions

**Conceptualization:** Adil Aslam Mir, Kimberlee Jane Kearfott, Fatih Vehbi Çelebi, Muhammad Rafique.

**Data curation:** Kimberlee Jane Kearfott, Fatih Vehbi Çelebi, Muhammad Rafique.

**Formal analysis:** Adil Aslam Mir, Kimberlee Jane Kearfott, Fatih Vehbi Çelebi, Muhammad Rafique.

**Investigation:** Adil Aslam Mir, Fatih Vehbi Çelebi, Muhammad Rafique.

**Methodology:** Adil Aslam Mir, Kimberlee Jane Kearfott, Fatih Vehbi Çelebi, Muhammad Rafique.

**Project administration:** Fatih Vehbi Çelebi, Muhammad Rafique.

**Resources:** Muhammad Rafique.

**Software:** Adil Aslam Mir.

**Supervision:** Fatih Vehbi Çelebi, Muhammad Rafique.

**Validation:** Adil Aslam Mir, Fatih Vehbi Çelebi, Muhammad Rafique.

**Visualization:** Adil Aslam Mir, Kimberlee Jane Kearfott, Fatih Vehbi Çelebi, Muhammad Rafique.

**Writing – original draft:** Adil Aslam Mir.

**Writing – review & editing:** Kimberlee Jane Kearfott, Fatih Vehbi Çelebi, Muhammad Rafique.

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
