## [Decision Letter · Decision Letter 0]

13 Jul 2021

PONE-D-21-18941

Imputation by feature importance (IBFI): A methodology to envelop machine learning method for imputing missing patterns in time series data

PLOS ONE

Dear Dr. Rafique,

Thank you for submitting your manuscript to PLOS ONE. After careful consideration, we feel that it has merit but does not fully meet PLOS ONE’s publication criteria as it currently stands. Therefore, we invite you to submit a revised version of the manuscript that addresses the points raised during the review process.

We look forward to receiving your revised manuscript.

Kind regards,

Shamsuddin Shahid

Academic Editor

PLOS ONE

Journal Requirements:

2. Please provide the full raw data set and any relevant code as supplemental files.

"Muhammad Rafique 

Grant No: 6453/AJK/NRPU/R&D/HEC/2016 under NRPU scheme to principal investigator MR. www.hec.gov.pk

The funders had no role in study design, data collection and analysis, decision to publish, or preparation of the manuscript". 

We note that one or more of the authors is affiliated with the funding organization, indicating the funder may have had some role in the design, data collection, analysis or preparation of your manuscript for publication; in other words, the funder played an indirect role through the participation of the co-authors. If the funding organization did not play a role in the study design, data collection and analysis, decision to publish, or preparation of the manuscript and only provided financial support in the form of authors' salaries and/or research materials, please do the following:

a. Review your statements relating to the author contributions, and ensure you have specifically and accurately indicated the role(s) that these authors had in your study. These amendments should be made in the online form.

b. Confirm in your cover letter that you agree with the following statement, and we will change the online submission form on your behalf: 

“The funder provided support in the form of salaries for authors [insert relevant initials], but did not have any additional role in the study design, data collection and analysis, decision to publish, or preparation of the manuscript. The specific roles of these authors are articulated in the ‘author contributions’ section.

Reviewers' comments:

Reviewer's Responses to Questions

**Comments to the Author**

1. Is the manuscript technically sound, and do the data support the conclusions?

Reviewer #1: Yes

Reviewer #2: Yes

Reviewer #3: Yes

2. Has the statistical analysis been performed appropriately and rigorously? 

Reviewer #1: Yes

Reviewer #2: Yes

Reviewer #3: No

3. Have the authors made all data underlying the findings in their manuscript fully available?

Reviewer #1: No

Reviewer #2: Yes

Reviewer #3: No

4. Is the manuscript presented in an intelligible fashion and written in standard English?

Reviewer #1: No

Reviewer #2: Yes

Reviewer #3: Yes

5. Review Comments to the Author

Reviewer #1: This study presents a new methodology to impute or gap-fill missing data. The methodology broadly leverages the strength of correlations among sampled variables. It seems to be generic enough and can be utilized with any learning algorithm. Although the proposed methodology seems to be free from any technical flaws, I could not follow all aspects of the work. In particular, the model reusability component of the methodology is not clear to me and can benefit with more clarification.

I also have a few more comments that are geared towards clarifying some aspects of the presentation. Therefore, I recommend that the manuscript be subject to moderate to major revisions. There are also places where the language of the manuscript includes errors. I have marked a few lines in the comments which especially caught the eye. I recommend the authors do a thorough reading of the manuscript before resubmitting.

Detailed comments:

1. The abstract and introduction talk about the importance of soil radon gas concentration (SRGC). Following this, the expectation is that the work will focus on imputation of SRGC data. However, the methodology is tested on imputation of five different variables: Radon, Thoron, Temperature, Relative Humidity, and Pressure. Therefore, the introduction should be revised or expanded to motivate the importance of all five variables considered in the study, and not just radon.

2. Introduction, line 68: “with the exact mechanism being unimportant for classification”. It is not clear what “classification” is being referred to here.

3. Introduction, line 72: “The nature of absent data, or missingness, can be classified in three ways”. I suggest to cite some references for this statement. E.g., works by Rubin [1] and Buuren [2].

4. Introduction, lines 79-93: The authors review several imputation approaches. However, the review lacks completeness. First, it would be useful if the various approaches that are reviewed by the authors can be explained in a sentence or two. Second, while the cons of simple approaches are clearly outlined, other approaches are not adequately discussed. For instance, the authors mention the benefits of multiple imputation and stochastic regression methods, but their shortcomings are not mentioned. Third, the authors mention that machine learning tends to outperform traditional statistical methods. Which of the methods reviewed in the literature fall under the realm of traditional statistical methods? Finally, given the methodology utilizes correlations among predictor variables to model a response variable, it is recommended to briefly review some relevant recent work. For instance, Mital et al. [3] and Sahu et al. [4] investigated the impact of selecting highly correlated input features for modeling/imputing a response variable.

5. Materials and Methods; Instrumentation and location:

I suggest that the authors provide a figure showing the location of the data used in this study. While it is not strictly necessary, it helps make the presentation more complete.

6. Materials and Methods, lines 128-129: “The missing values are introduced into the dataset artificially by the R package entitled mice”. I recommend that a brief description of how the missing values were inserted should be provided. Simply stating that the missing values were inserted using a package sounds opaque and is not sufficient. How does the package insert values that are consistent with the three different missingness patterns? What are the mechanics of inserting those missing values?

7. Material and Methods, lines 189-191:

For kernel density estimation, what kernel is used? Is it the normal kernel or something else? Please clarify in the text.

8. Hot deck imputation method, lines 194-202:

The description of the method is loaded with jargon that may not make much sense to a reader unfamiliar with this method. For instance, what is a “responding unit”? What is a “practical response”? I suggest that the description be re-worded and made more accessible.

9. Predictive mean matching imputation method, lines 204-216:

Again, the method description is not clear. The context of sentences on lines 210-213 is not clear to an uninitiated reader (such as myself). Furthermore, on lines 215-216, the authors list the parameters used in the method. It is not clear what these parameters mean and how these values were chosen.

10. Pseudo code, line 226:

The current presentation of the pseudo code seems very complex and could benefit with simplification. In particular, it seems to use the syntax and functions used in programming language R. I recommend that the code be revised to make it more readable for someone who is not familiar with R.

11. Line 249:

What is a “sample” and a “value” in the context of this work? It seems that one sample refers to a measurement of all five attributes (values). The terminology should be clarified and made consistent across the manuscript. For instance, the terms “values” and “attributes” have been used interchangeably.

12. Lines 268-270: Please rephrase to correct the grammar.

13. Lines 291-303: Please rephrase to correct the grammar.

14. Lines 296-303: “The proposed methodology uses model reusability”. The description of model reusability is not clear to me. Specifically, in the scenario described in lines 298-303, it is not clear to me why F1 needs to be trained again using F2 and F3 during subsequent iterations.

15. Concerning feature importance: Is feature or variable importance quantified using correlations? If so, please clarify.

16. Variables RN, TH, TC, RH and PR: Please define these abbreviations.

17. Fig 4: Overall, I really like this figure. It helps the reader to understand all the results qualitatively. However, I did not follow how the normalization was done quantitatively. Please clarify by either rephrasing or perhaps giving an example of normalization. There is also one minor typo in the y-axis labels (“Mean: R<s:e”).

18. Fig 6: I did not understand the results in this figure since the concept of model reusability was not clear to me (see comment 14 above). Furthermore, the terms “model hit rate” and “model creation” have not been defined.

19. Lines 364-368: Please rephrase to correct the grammar.

20: Concerning “rejection threshold” or “rejection count”: How do the authors pick an appropriate value of the rejection threshold? Why did the authors pick a value of 3? If the number of attributes for a sample is 5, I would assume that a value of 4 may also work. Also, I would recommend keeping the terminology consistent to avoid ambiguity, i.e., use either the phrase “rejection threshold” or “rejection count”.

21: Concerning “Keywords”: I suggest the authors revise the keywords for the manuscript. The “Naïve Bayes” classifier and “Random Forests” are not used in this work and should not be used as keywords.

References:

1. Rubin DB. Inference and missing data. Biometrika. 1976;63: 581–592.

2. Buuren S van. Flexible imputation of missing data. Second edition. Boca Raton: CRC Press, Taylor & Francis Group; 2018.

3. Mital U, Dwivedi D, Brown JB, Faybishenko B, Painter SL, Steefel CI. Sequential Imputation of Missing Spatio-Temporal Precipitation Data Using Random Forests. Front Water. 2020;2: 20. doi:10.3389/frwa.2020.00020

4. Sahu RK, Müller J, Park J, Varadharajan C, Arora B, Faybishenko B, et al. Impact of Input Feature Selection on Groundwater Level Prediction From a Multi-Layer Perceptron Neural Network. Front Water. 2020;2: 573034. doi:10.3389/frwa.2020.573034</s:e”).

Reviewer #2: Abstract: The authors need to revise it to highlight the problem, findings and novelty of their work.

Introduction: The literature review in this section needs to be updated with recent published works relevant to the study. The authors should improve the problem statement and highlight the objectives of the study clearly and mention the main contribution in the study.

Material and Methods: More explanation about the importance of the proposed model to solve the current problem.

Results and discussion: The authors are encouraged to add more explanations to the findings and justify it clearly.

Conclusion: The authors are advised to re-write this section to justify the findings and suggest future work to be carried in terms of deploying the models or proposing way to enhance it.

References: The authors missed out recent references related to their works which they are encouraged to included in their revised version

Reviewer #3: Review Report of Imputation by feature importance (IBFI): A methodology to envelop machine learning method for imputing missing patterns in time series data

Although the paper may present a new methodology, it is badly written. My recommendation is “major revision”.

The comments:

The author mentioned in the abstract, introduction, and conclusion that they have used XGBoost, however; the XGBoost was not mentioned once in the methodology section. Yes, it is there in Figure 2 but it was not found in the text. Thus, it is not clear the role that XGBoost played in the proposed framework.

I don’t understand why the authors employed several statistical metrics which measure the same characteristics. For example, RMSE and MSE, and RMSLE are somehow the same. In a matter of fact, the author showed that RMSE and MSE have several disadvantages. The authors should use only one of them like RMSLE and remove the remaining. The others make the paper longer for no added information. The same goes for MAPE and PB, I encourage the authors to pick only one of them. This will reduce the paper length and will give more focus to the new framework.

Also, I feel it is not fair to compare the new framework to conventional data filling techniques. Obviously, the new framework will be better. I encourage the authors to add a random forest or support vector machine model to the comparison which may increase the work strength.

A brief description of the data should be given. Yes, it may be described elsewhere. I think a very brief statistical description of the data itself is also required here.

The introduction is badly written. It doesn’t follow a line of ideas. Many ideas are repeated here and there in the introduction section. Please, review it once more.

6. PLOS authors have the option to publish the peer review history of their article (what does this mean?). If published, this will include your full peer review and any attached files.

Reviewer #1: No

Reviewer #2: No

Reviewer #3: **Yes: **Mohamed Salem Nashwan

---

## [Author Response · Author response to Decision Letter 0]

27 Aug 2021

Dear Editor and Editorial Staff,

PLOS ONE

We are pleased to inform you that we have revised the manuscript in the light of reviewers’ comments. The anonymous reviewers’ recommendations were extremely useful and we have addressed all of their recommendations in the revised manuscript. Please see below for responses to each individual comment. 

Reviewer #1:

Comment The abstract and introduction talk about the importance of soil radon gas concentration (SRGC). Following this, the expectation is that the work will focus on imputation of SRGC data. However, the methodology is tested on imputation of five different variables: Radon, Thoron, Temperature, Relative Humidity, and Pressure. Therefore, the introduction should be revised or expanded to motivate the importance of all five variables considered in the study, and not just radon.

Response We have incorporated necessary changes in introduction in order to address reviewers concern. As meteorological parameters have great influence on radon emission dynamics that can be used as a precursor for earthquake, reference material is added in the introduction to highlight the importance of variables other than radon along with the methodology to impute all these variables. 

Comment Introduction, line 68: “with the exact mechanism being unimportant for classification”. It is not clear what “classification” is being referred to here.

Response We have revised the line as pointed out by respected reviewer. The term “classification” here refers to the categories of missingness such as missing completely at random, missing not at random and missing at random. The incomplete or missing data can be classified according to the mechanism through which it generates such as MCAR if missing value is generated by human or machine error. If the cause of missingness is related to observed data but not the missing data, it is classified as MAR while if it is missing because of other observed variables or some hypothetical value, it is MNAR.

Comment Introduction, line 72: “The nature of absent data, or missingness, can be classified in three ways”. I suggest citing some references for this statement. E.g., works by Rubin [1] and Buuren [2].

Response The references provided by the respected reviewers have been added in the manuscript. 

Comment Introduction, lines 79-93: The authors review several imputation approaches. However, the review lacks completeness. First, it would be useful if the various approaches that are reviewed by the authors can be explained in a sentence or two. Second, while the cons of simple approaches are clearly outlined, other approaches are not adequately discussed. For instance, the authors mention the benefits of multiple imputation and stochastic regression methods, but their shortcomings are not mentioned. Third, the authors mention that machine learning tends to outperform traditional statistical methods. Which of the methods reviewed in the literature fall under the realm of traditional statistical methods? Finally, given the methodology utilizes correlations among predictor variables to model a response variable, it is recommended to briefly review some relevant recent work. For instance, Mital et al. [3] and Sahu et al. [4] investigated the impact of selecting highly correlated input features for modeling/imputing a response variable.

Response Agreed. In addition to the simple approaches towards imputation tasks, the discussions about other imputation methods such as multiple imputations have been provided. The shortcomings of imputation methods are also provided in the manuscript to address the reviewer concern. We have also reviewed and added material regarding several machine learning methods for imputing missing data such as sequential imputation using Random Forest for imputing missing values in spatio-temporally daily time series precipitation records. 

Comment Materials and Methods; Instrumentation and location:

I suggest that the authors provide a figure showing the location of the data used in this study. While it is not strictly necessary, it helps make the presentation more complete.

Response Agreed. We have added the figure regarding instrumentation and location in the manuscript with caption “Fig 1. Soil radon measuring station located inside 150 km from the epicenter of the strongest earthquake since 1900 with the latitude, longitude of 34.396210 and 73.473470 respectively”.

Comment Materials and Methods, lines 128-129: “The missing values are introduced into the dataset artificially by the R package entitled mice”. I recommend that a brief description of how the missing values were inserted should be provided. Simply stating that the missing values were inserted using a package sounds opaque and is not sufficient. How does the package insert values that are consistent with the three different missingness patterns? What are the mechanics of inserting those missing values?

Response In order to address the reviewer concern, we have incorporated detailed mechanics of inserting missing values in the dataset for three different missingness scenarios such as MCAR, MAR and MNAR. The core idea to introduce the missing values, using R package “mice”, in the multivariate dataset lies in the missing patterns which are the mixture of variables with missing values and variables with available values. The missing patterns with its frequency are shown in Fig 3. The complete dataset is divided into k subsets randomly based upon k missing data patterns. The subset size depends upon the frequency vector which is the frequency of the certain pattern to be missing the complete dataset. The data rows in the subsets are considered to be a candidate for missing is based upon several factor such as missingness mechanism (MCAR, MNAR and MAR). In MCAR scenarios, all the data rows in the subsets have the equal probability of being missing while in MNAR and MAR scenarios, the so-called weighted sum scores are computed. More simply put, the weighted sum scores are the outcome of a linear regression equation and these scores provides basis for candidates data rows to be missing or not. Finally, the data rows in the subsets are made missing or incomplete according to the missing data pattern along with its probability for being missing. After the introduction of missing values, these subsets are merged to make incomplete dataset having missing values in different data rows.

Comment Material and Methods, lines 189-191:

For kernel density estimation, what kernel is used? Is it the normal kernel or something else? Please clarify in the text.

Response As the data used in this study is continuous time series data, kernel density estimation is used to produce a continuous estimate of probability density function. The point at which that function reaches its maximum is considered the mode. For kernel density estimation, R package “stats” [69] is used. The function “density ()” with its default parameters are used to calculate the kernel density estimate. The function “density.default ()” uses the algorithm that first use a regular grid of at least 512 points to disperse the mass of the empirical distribution function. The fast Fourier transform is used to convolve this approximation along with the discretized version of the kernel such as Gaussian. Finally, the density at specified points is evaluated using linear approximation.

Comment Hot deck imputation method, lines 194-202:

The description of the method is loaded with jargon that may not make much sense to a reader unfamiliar with this method. For instance, what is a “responding unit”? What is a “practical response”? I suggest that the description be re-worded and made more accessible.

Response Agreed. We have incorporated necessary changes in manuscript in order to address reviewers concern. We have re-worded the text to make it more accessible and easy to understand even for unfamiliar reader. 

Comment Predictive mean matching imputation method, lines 204-216:

Again, the method description is not clear. The context of sentences on lines 210-213 is not clear to an uninitiated reader (such as myself). Furthermore, on lines 215-216, the authors list the parameters used in the method. It is not clear what these parameters mean and how these values were chosen.

Response Agreed. We have incorporated necessary changes in manuscript in order to address reviewers concern. We have re-worded the text to make it more accessible and easy to understand.

Comment Pseudo code, line 226:

The current presentation of the pseudo code seems very complex and could benefit with simplification. In particular, it seems to use the syntax and functions used in programming language R. I recommend that the code be revised to make it more readable for someone who is not familiar with R.

Response Agreed. We have rewrite the pseudo code to somewhat algorithmic style and easy to interpret for all type of audiences even if they do not have the knowledge and syntax of R programming language. 

Comment Line 249:

What is a “sample” and a “value” in the context of this work? It seems that one sample refers to a measurement of all five attributes (values). The terminology should be clarified and made consistent across the manuscript. For instance, the terms “values” and “attributes” have been used interchangeably.

Response Agreed with the respected reviewer. The sample is the measurement of all five attributes or variables while value is the single measurement for different attributes such as temperature has the value of 38.5oC. In order to address the reviewer concern, we have made the terms consistent throughout the script in the manuscript.

Comment Lines 268-270: Please rephrase to correct the grammar.

Response We have incorporated necessary changes in manuscript in order to address reviewers concern.

Comment Lines 291-303: Please rephrase to correct the grammar.

Response We have incorporated necessary changes in manuscript in order to address reviewers concern.

Comment Lines 296-303: “The proposed methodology uses model reusability”. The description of model reusability is not clear to me. Specifically, in the scenario described in lines 298-303, it is not clear to me why F1 needs to be trained again using F2 and F3 during subsequent iterations.

Response As the proposed methodology imputes the attributes or variables using feature importance matrix, the sequence of imputation of values for different samples differs for each other. The whole procedure is scanning the impure dataset with respect to the rejection threshold. Consider sample 1, the missing pattern have attribute F1 and F5 having missing value in it and sample 100 have missing values at attributes F1, F2 and F5. For sample 1, F2, F3 and F4 have values available to predict the appropriate value for F1 and F5 while F3 and F4 have only available values to predict for F1, F2 and F5. During scan 1 and iteration 1, the feature importance matrix directs the algorithm to predict the value for F1 using F2, F3 and F4 by taking F1 as a response attribute and F2, F3 and F4 as predictor attributes. After predicting and imputing the value of F1, the model should store as M1234 for future patterns. During the same scan when reached at sample no. 100, feature importance matrix directs the algorithm to impute the value of F2 which is missing by taking F2 as a response attribute while F3 and F4 as predictor attributes. After predicting and imputing the value of F2, the sample no. 100 has now an available value of F2 to predict other attributes. Now consider second iteration and scan no. 100, the feature importance matrix directs the algorithm to impute the value of F1 first instead of F5. Now, F2, F3 and F4 are available values to predict the value of F1. Instead of fitting another model to predict the value of F1, the previously generated M1234 model is reused and values of F2, F3 and F4 are passed to that model in order to predict the value of F1. This reusability enhances the performance of proposed methodology and reduces the computation to fit another machine learning model. 

Comment Concerning feature importance: Is feature or variable importance quantified using correlations? If so, please clarify.

Response The feature importance matrix is computed by using R package “randomForest”. For each feature in the dataset by taking it as response feature and others as predictor, the feature importance of other features for predicting that response is computed. The features are arranged as per their importance value from highest to lowest. The clarification of this step is incorporated in the manuscript in order to address the reviewer concern.

Comment Variables RN, TH, TC, RH and PR: Please define these abbreviations.

Response RN, TH, TC, RH and PR stands for Radon, Thoron, Temperature, Relative Humidity and Pressure. We have also incorporated their full forms in the revised manuscript.

Comment Fig 4: Overall, I really like this figure. It helps the reader to understand all the results qualitatively. However, I did not follow how the normalization was done quantitatively. Please clarify by either rephrasing or perhaps giving an example of normalization. There is also one minor typo in the y-axis labels (“Mean: R

18. Fig 6: I did not understand the results in this figure since the concept of model reusability was not clear to me (see comment 14 above). Furthermore, the terms “model hit rate” and “model creation” have not been defined.

Response Agreed. As far as figure 4 is of concern, the average statistics are calculated for all variables across different missingness scenarios such as MCAR, MNAR and MAR. Firstly, the average of each performance metric among all imputation methods is calculated for the entire variables across different missingness scenarios with their associated missingness percentages. Secondly, each value in different missingness scenarios such as MCAR 10% is normalized with respect to different performance metric by dividing with the corresponding average value calculated in step 1. Now we have the normalized value of each performance metric across different missingness scenarios for different missingness percentages. Thirdly, the averages of all the variables are calculated with respect to imputation method. Finally, the average statistics of all the performance metrics across all variables are presented in Figure 4.

 We have corrected the typo in y-axis label to address the reviewer concern. 

The detailed answer for model reusability is provided in the comment above. The “model hit rate” and “model creation” is now defined in the manuscript. Model hit reflects the previous model reusability. While imputing each value, before model creation, the current formulated model directed by feature importance matrix is searched in the model list. If it is found, the already fitted model should serve to predict the missing value at this stage. In the case when the model hit is not occurred, the new model is fitted for current formulation and the fitted model is added to the model list which may use for further formulations. 

Comment Lines 364-368: Please rephrase to correct the grammar.

Response We have incorporated necessary changes in manuscript in order to address reviewers concern.

Comment Concerning “rejection threshold” or “rejection count”: How do the authors pick an appropriate value of the rejection threshold? Why did the authors pick a value of 3? If the number of attributes for a sample is 5, I would assume that a value of 4 may also work. Also, I would recommend keeping the terminology consistent to avoid ambiguity, i.e., use either the phrase “rejection threshold” or “rejection count”.

Response Agreed. In order to avoid ambiguity, we have reworded the phrase to “rejection threshold” instead of “rejection count” in the manuscript. The rejection threshold controls the extent to which the numbers of attribute values are missing in different samples. If there is an increasing number of missing values in different samples, there is chance that imputing the values may result in contamination of those samples and results in poor analyses or biased result on further experimentation using that imputed dataset. Although, the rejection threshold can be 4 and it works but in our case, we have total of 5 attributes. The value of 3 is selected based upon the assumption that at least there should be two attributes available that can serve as the predictor attributes to get more accurate results. 

Comment Concerning “Keywords”: I suggest the authors revise the keywords for the manuscript. The “Naïve Bayes” classifier and “Random Forests” are not used in this work and should not be used as keywords.

Response Agreed. We have incorporated necessary changes in manuscript in order to address reviewers concern.

Reviewer #2:

Comment Abstract: The authors need to revise it to highlight the problem, findings and novelty of their work.

Response We have revisited all the concerns that were mentioned by the respected reviewer and also responded in detailed form in the revised manuscript. We have provided the more detailed description of the scenarios where the proposed methodology can be beneficial. Basically, this method is more useful in the scenarios where more than one missing values occurs in different samples. In order to predict the missing value for that sample, only a single model based upon all the other predictor variables is not enough. This is due to the fact that the prediction of those missing values in certain samples requires all the available values to predict it using any machine learning method. Using this method, one can impute the missing values in whole dataset automatically using any base machine learning algorithm without taking care of how much missing values occurs in different samples. The number of imputations in different samples is controlled by the user of this methodology.

Comment Introduction: The literature review in this section needs to be updated with recent published works relevant to the study. The authors should improve the problem statement and highlight the objectives of the study clearly and mention the main contribution in the study.

Response Agreed. This is also pointed out by other respected reviewers also. To address this concern, we have restructured the introduction part by providing more related work for missing data imputation. We have also provided cons of other imputation methods used in this paper. 

Comment Material and Methods: More explanation about the importance of the proposed model to solve the current problem.

Response We have revisited all the concerns that were mentioned by the respected reviewer and also responded in detailed form in the revised manuscript. We have described the parameters and also provided the detailed information of proposed methodology as well as other imputation methods used in this study. We have provided deeper look towards model reusability and its effectiveness in the imputation process regarding proposed methodology.

Comment Results and discussion: The authors are encouraged to add more explanations to the findings and justify it clearly.

Response We have revisited all the concerns that were mentioned by the respected reviewer and also responded in detailed form in the revised manuscript.

Comment Conclusion: The authors are advised to re-write this section to justify the findings and suggest future work to be carried in terms of deploying the models or proposing way to enhance it.

Response We have rewritten this section as per directions of the respected reviewer. We have also provided future work that can be carried out using proposed methodology. The scenarios where it can be beneficial for missing values imputation are also discussed.

Comment References: The authors missed out recent references related to their works which they are encouraged to included in their revised version

Response We have revisited all the concerns that were mentioned by the respected reviewer and also responded in detailed form in the revised manuscript. The related references are added to the revised manuscript.

Reviewer #3:

Comment The author mentioned in the abstract, introduction, and conclusion that they have used XGBoost, however; the XGBoost was not mentioned once in the methodology section. Yes, it is there in Figure 2 but it was not found in the text. Thus, it is not clear the role that XGBoost played in the proposed framework.

Response We have revisited all the concerns that were mentioned by the respected reviewer and also responded in detailed form in the revised manuscript. We have provided the role of XGBoost for using it as base learning algorithm in proposed methodology. Apart from XGBoost, using this methodology by employing any base learning algorithm of our choice, one can impute the missing values in whole dataset automatically without taking care of how much missing values occurs in different samples. The number of imputations in different samples is controlled by the user of this methodology.

Comment I don’t understand why the authors employed several statistical metrics which measure the same characteristics. For example, RMSE and MSE, and RMSLE are somehow the same. In a matter of fact, the author showed that RMSE and MSE have several disadvantages. The authors should use only one of them like RMSLE and remove the remaining. The others make the paper longer for no added information. The same goes for MAPE and PB; I encourage the authors to pick only one of them. This will reduce the paper length and will give more focus to the new framework.

Response The reason behind the use of multiple metrics for performance evaluation because the imputation of missing values is carried out for all the variables (radon, thoron, temperature, relative humidity and pressure) in the soil gas radon concentration time series dataset and all the variables has different type of scales. The radon concentration time series with the minimum of 13743 to maximum of 28085 Bq/m3 while temperature time series ranges from 4 to 42.5 0C in the dataset. The resultant prediction error for radon and thoron is much larger when considering against temperature and relative humidity. RMSLE doesn’t penalize large errors. It is usually used when we don’t want to influence the results if there are large errors. RMSLE penalize lower errors. When actual and predicted values are low, RMSE & RMSLE are usually same. MAPE is used to account for attributes such as radon and thoron because of its less biasness towards higher values. Finally, the PB is used to account for overestimation or underestimation bias. The PB tells more insights apart from RMSE, RMSLE, MSE and MAPE. Thus, instead of relying on a single performance evaluation metric, multiple evaluation metric or loss functions are used to accommodate for all the variables in the dataset. Moreover, the resultant values for all the performance evaluation metrics are provided as supplementary material with the manuscript. The normalized values of these performance evaluation metrics are provided in the form of figures in the manuscript. 

Comment Also, I feel it is not fair to compare the new framework to conventional data filling techniques. Obviously, the new framework will be better. I encourage the authors to add a random forest or support vector machine model to the comparison which may increase the work strength.

Response Agreed with the respected reviewer. We have compared the new framework with the conventional data filling techniques. The basic idea behind using the conventional data techniques lies upon the fact that we have introduced the missing values in all the five variables. The introduction of missing values in five variables leads to different patterns of missingness and multiple variables ranges from 1 to the number of variables may be missing per sample. In order to predict the missing value for those samples having more than one missing values, only a single raw model (such as random forest and support vector machines) based upon all the other predictor variables is not enough. This is due to the fact that the prediction of those missing values in certain samples requires all the available values to predict it using any machine learning method. Thus, we have a need of robust iterated methodology that can envelop these basic machine learning models to get adapted as per missingness scenarios in different samples. This methodology make the basic machine learning method to be able to impute more than one missing values in different samples automatically by using feature importance matrix without taking care of how much missing values occurs in different samples. The number of imputations in different samples is controlled by the user of this methodology.

Comment A brief description of the data should be given. Yes, it may be described elsewhere. I think a very brief statistical description of the data itself is also required here.

The introduction is badly written. It doesn’t follow a line of ideas. Many ideas are repeated here and there in the introduction section. Please, review it once more.

Response We have revisited all the concerns that were mentioned by the respected reviewer and also responded in detailed form in the revised manuscript. We have provided the statistical details of the data in the form of table in the revised manuscript. The introduction part is restructured in the revised manuscript and maintains the flow of ideas as well as recent work regarding imputation of missing data is also provided. 

Kind Regards

Prof. Dr. M. Rafique

---

## [Decision Letter · Decision Letter 1]

3 Nov 2021

PONE-D-21-18941R1Imputation by feature importance (IBFI): A methodology to envelop machine learning method for imputing missing patterns in time series dataPLOS ONE

Dear Dr. Rafique,

Thank you for submitting your manuscript to PLOS ONE. After careful consideration, we feel that it has merit but does not fully meet PLOS ONE’s publication criteria as it currently stands. Therefore, we invite you to submit a revised version of the manuscript that addresses the points raised during the review process.

We look forward to receiving your revised manuscript.

Kind regards,

Shamsuddin Shahid

Academic Editor

PLOS ONE

Journal Requirements:

Reviewers' comments:

Reviewer's Responses to Questions

**Comments to the Author**

1. If the authors have adequately addressed your comments raised in a previous round of review and you feel that this manuscript is now acceptable for publication, you may indicate that here to bypass the “Comments to the Author” section, enter your conflict of interest statement in the “Confidential to Editor” section, and submit your "Accept" recommendation.

Reviewer #1: (No Response)

2. Is the manuscript technically sound, and do the data support the conclusions?

Reviewer #1: Yes

3. Has the statistical analysis been performed appropriately and rigorously? 

Reviewer #1: Yes

4. Have the authors made all data underlying the findings in their manuscript fully available?

Reviewer #1: No

5. Is the manuscript presented in an intelligible fashion and written in standard English?

Reviewer #1: No

6. Review Comments to the Author

Reviewer #1: The authors have presented a revised version of their manuscript. Overall, I do not think the manuscript is ready for acceptance yet. While I do not question the methodology and the results, the presentation and text need to be thoroughly proofread for clarity and grammar before the manuscript can be considered to be of publication quality. Although the authors have sought to address all my comments, the responses lack clarity and, at times, were accompanied by poor grammar in the manuscript. I had to re-read the responses multiple times to understand them and had to eventually guess an explanation that made the most sense. Care needs to be taken to ensure that the language used in the manuscript is precise. Given that PLOS ONE does not copyedit accepted manuscripts, this is very important.

1. Please define SRGC in the introduction when it is mentioned for the first time.

2. Response 2, lines 93-94: Please grammar-check

3. Response 4: please grammar-check the description of various imputation methods that have been added to the introduction

4. Response 6: description of missingness mechanisms lacks clarity; please proofread

5. Response 8: there is reference to PMM in the description of hot-deck imputation. This adds confusion since PMM is described in the next section. Further, the comment asked the authors to rephrase the description to remove jargon. However, the description is now too detailed which ultimately still falls short of explaining the method adequately (e.g, how is the donor pool picked?). An effort needs to be made to keep the description short by providing only the relevant information.

6. Response 9: this comment has not been addressed adequately. The mechanics of PMM have not been explained, nor have the parameters been described.

7. Response 10: The pseudo-code is still difficult to follow.

8. Response 14: The description of model reusability needs to be improved for readability, and then inserted in the manuscript.

Also, while I agree that reusability reduces computation time, I am not sure how it results in more accurate predictions. Finally, I assume that “pure data” PD is used to create models. If so, that should be clarified. Presently, the purpose of PD has not been explained explicitly anywhere in the text.

9. Response 17: In Figure 7 of the revision, what is the measurement number? You mentioned "scans" and "iterations" in response 14. How does it relate to that?

10. Response 19: Please state your assumption justifying the use of a rejection threshold of 3 in the manuscript.

11. It seems that PLOS data policy requires data underlying the findings to be fully available, which includes the data points behind the summary statistics. I only see the summary statistics in the supplement.

7. PLOS authors have the option to publish the peer review history of their article (what does this mean?). If published, this will include your full peer review and any attached files.

Reviewer #1: No

---

## [Author Response · Author response to Decision Letter 1]

20 Nov 2021

Dear Editor and Editorial Staff,

PLOS ONE

We are pleased to inform you that we have revised the manuscript in light of reviewers’ comments. The anonymous reviewers’ recommendations were extremely useful and we have addressed all of their recommendations in the revised manuscript. Please see below for responses to each comment. 

Reviewer #1:

Comment Please define SRGC in the introduction when it is mentioned for the first time.

Response We have incorporated necessary changes in the introduction to address reviewers concern.

Comment Response 2, lines 93-94: Please grammar-check

Response We have proofread as well as grammatically checked the lines 93-94 in the revised manuscript.

Comment Response 4: please grammar-check the description of various imputation methods that have been added to the introduction

Response We have proofread as well as grammatically checked the description of various imputation methods in the revised manuscript.

Comment Response 6: description of missingness mechanisms lacks clarity; please proofread

Response To address the reviewer's concern, we have restructured and clarified the concept in a more precise and easy way. 

Comment Response 8: there is reference to PMM in the description of hot-deck imputation. This adds confusion since PMM is described in the next section. Further, the comment asked the authors to rephrase the description to remove jargon. However, the description is now too detailed which ultimately still falls short of explaining the method adequately (e.g, how is the donor pool picked?). An effort needs to be made to keep the description short by providing only the relevant information.

Response Agreed with the findings of the respected reviewer. The reference in the hot-deck imputation section adds confusion when reading the description of the PMM imputation method. To address the concern, we have added relevant details of each imputation method and rephrased and proofread as well. 

Comment Response 9: this comment has not been addressed adequately. The mechanics of PMM have not been explained, nor have the parameters been described.

Response To address this concern, we have explained the mechanics of PMM and its parameters in a more precise and easy way so that the concept can be easily figured out. 

Comment Response 10: The pseudo-code is still difficult to follow.

Response To address the reviewer's concern, we have added comments and restructured some statements in the pseudo-code in a way that can be easily followed. 

Comment Response 14: The description of model reusability needs to be improved for readability, and then inserted in the manuscript.

Also, while I agree that reusability reduces computation time, I am not sure how it results in more accurate predictions. Finally, I assume that “pure data” PD is used to create models. If so, that should be clarified. Presently, the purpose of PD has not been explained explicitly anywhere in the text.

Response The proposed methodology imputes the attributes or variables using a feature importance matrix. The sequence of imputation of values for the missing features in different samples differs from each other. The whole procedure is scanning the impure dataset as per the rejection threshold. Consider data row 1, the features F1 and F5 having a missing value while data row 100 have missing values at features F1, F2, and F5. Data row 1 has F2, F3, and F4 feature values available to predict the value of F1 and F5 whilst F3 and F4 have only available feature values to predict the missing value of F1, F2, and F5. During scan 1 and iteration 1, the feature importance matrix directs the algorithm to predict the missing value of F1 using the model fitted on features F2, F3, and F4 by considering F1 as a response and F2, F3, and F4 as predictor features. After predicting and imputing the missing value of F1, the model should store as M1234 for future patterns. Through the same scan when reached at data row 100, the feature importance matrix directs the algorithm to impute the value of F2 which is missing by taking F2 as a response attribute while F3 and F4 as predictor attributes. After predicting and imputing the missing value of F2, the data row 100 has now an available value of F2 to predict other features. Now consider the second iteration and scan 100, the feature importance matrix directs the algorithm to impute the value of F1 first instead of F5. Currently, F2, F3, and F4 are available values to predict the missing value of F1. Instead of fitting another model to predict the missing value of F1, the previously generated M1234 model is reused and values of F2, F3, and F4 are passed to that model to predict the missing value of F1. The reusability of the already fitted model to impute the similar missing patterns enhance the performance and reduces the time and space requirements. 

 The accuracy of predicting the missing value for different features is based upon the feature importance matrix and runtime decision of choosing the best available features for training while model reusability results in reducing the computation time. 

 Agreed with the respected reviewer. The pure data (PD) has all the feature values available to fit a machine learning model. To address this concern, we have explicitly explained the usage of PD in imputation by feature importance (IBFI).

 Comment Response 17: In Figure 7 of the revision, what is the measurement number? You mentioned "scans" and "iterations" in response 14. How does it relate to that?

Response In IBFI, “scan” traverses the whole impure data by scanning each data row to check for missing patterns and selection of best available features for training. On the other hand, “iteration” consists of one full scan of the impure data. During each iteration, the number of missing features per data row gets reduced by one feature. 

 In figure 7, the measurement number represents the data rows handled for imputing missing patterns by IBFI during the subsequent scans and iterations.

Comment Response 19: Please state your assumption justifying the use of a rejection threshold of 3 in the manuscript.

Response If there is an increasing number of missing values in different data rows, there is the chance that imputing the values may result in contamination of those data rows and result in poor analyses or biased results on further experimentation using that imputed dataset. 

 The dataset used in this study has 5 features. The assumption behind the use of a rejection threshold of 3 is to ensure that at least 2 features are available for model training. The imputation of more than 3 out of 5 features may result in a relatively less accurate imputation of missing value because of the use of previously imputed value for the prediction of the other missing feature values.

Comment It seems that PLOS data policy requires data underlying the findings to be fully available, which includes the data points behind the summary statistics. I only see the summary statistics in the supplement.

Response The data that support the findings of this study are available from the corresponding author upon reasonable request.

Sincerely,

Prof. Dr. M. Rafique

---

## [Decision Letter · Decision Letter 2]

19 Dec 2021

Imputation by feature importance (IBFI): A methodology to envelop machine learning method for imputing missing patterns in time series data

PONE-D-21-18941R2

Dear Dr. Rafique,

We’re pleased to inform you that your manuscript has been judged scientifically suitable for publication and will be formally accepted for publication once it meets all outstanding technical requirements.

Kind regards,

Shamsuddin Shahid

Academic Editor

PLOS ONE

Additional Editor Comments (optional):

Reviewers' comments:

Reviewer's Responses to Questions

**Comments to the Author**

1. If the authors have adequately addressed your comments raised in a previous round of review and you feel that this manuscript is now acceptable for publication, you may indicate that here to bypass the “Comments to the Author” section, enter your conflict of interest statement in the “Confidential to Editor” section, and submit your "Accept" recommendation.

Reviewer #1: (No Response)

2. Is the manuscript technically sound, and do the data support the conclusions?

Reviewer #1: Yes

3. Has the statistical analysis been performed appropriately and rigorously? 

Reviewer #1: Yes

4. Have the authors made all data underlying the findings in their manuscript fully available?

Reviewer #1: No

5. Is the manuscript presented in an intelligible fashion and written in standard English?

Reviewer #1: Yes

6. Review Comments to the Author

Reviewer #1: All technical issues have been addressed.

Concerning the use of rejection threshold of 3, I suggest that the authors also state their justification explicitly in the manuscript. I also suggest that the authors explicitly state in the manuscript what they mean by "scans" and "iterations". This information was provided in the response document but I could not find it explicitly stated in the manuscript.

The other comment is about the lack of data availability. The authors have only provided the summary statistics and not the actual data points. I leave it up to the editor to adjudicate whether this satisfies the PLOS Data Policy.

7. PLOS authors have the option to publish the peer review history of their article (what does this mean?). If published, this will include your full peer review and any attached files.

Reviewer #1: No

---

## [Editor Report · Acceptance letter]

23 Dec 2021

PONE-D-21-18941R2 

Imputation by feature importance (IBFI): A methodology to envelop machine learning method for imputing missing patterns in time series data 

Dear Dr. Rafique:

I'm pleased to inform you that your manuscript has been deemed suitable for publication in PLOS ONE. Congratulations! Your manuscript is now with our production department. 

Kind regards, 

on behalf of

Dr. Shamsuddin Shahid 

Academic Editor

PLOS ONE